# Factors Associated with Fatigued Driving among Australian Truck Drivers: A Cross-Sectional Study

**DOI:** 10.3390/ijerph20032732

**Published:** 2023-02-03

**Authors:** Xinyi Ren, Elizabeth Pritchard, Caryn van Vreden, Sharon Newnam, Ross Iles, Ting Xia

**Affiliations:** 1Healthy Working Lives Research Group, School of Public Health and Preventive Medicine, Monash University, Melbourne, VIC 3004, Australia; 2Psychology and Counselling, Faculty of Health, Queensland University of Technology, Brisbane, QLD 4059, Australia

**Keywords:** truck drivers, fatigue, heavy vehicle drivers, fatigue driving, occupational risk factors

## Abstract

Background: Fatigued driving is one of the leading factors contributing to road crashes in the trucking industry. The nature of trucking, prolonged working time, and irregular sleep patterns can negatively impact drivers’ health and wellbeing. However, there is limited research in Australia investigating the impact of demographic, occupational, or lifestyle factors on fatigue among truck drivers. Objective: This cross-sectional study examines the role of demographic, occupational, lifestyle, and other health risk factors associated with fatigue among Australian truck drivers. Method: This study was part of a larger study that used a short online survey with a follow-up telephone survey to capture in-depth information on a wide range of determinants related to truck drivers’ physical and mental health outcomes. Fatigue was measured by three questions, including the frequency of fatigue, fatigue management training, and strategies used to combat fatigue. Multivariate regression analysis was used to determine the specific impact of demographics, occupational factors, lifestyle factors, and other health risk factors on fatigue. Results: In total, 332 drivers completed both the online and telephone surveys; 97% were male, representing drivers from broad age groups and professional experience. The odds of being in the high-risk fatigue group were nearly three times higher in drivers who worked 40–60 h compared to those who worked < 40 h. Poor sleep increased the odds of high-risk fatigue by seventimes (95% CI: 2.26–21.67, *p* = 0.001). Drivers who reported experiencing loneliness also had double the odds of being at high risk of fatigued driving. Conclusions: The increased risk of fatigue in truck drivers is associated with prolonged working hours, poor sleep, and social aspects such as loneliness. Further interventions seeking to reduce driver fatigue should consider the impact of work schedules, the availability of quality sleeping spaces, and the level of social connections.

## 1. Introduction

### Background

Road transport is the leading mode of freight transport in Australia, representing 75% of total freight transportation and is vital for the national economy, contributing 14.5% of the gross domestic product [1,2]. Australia has the most extensive distances to travel in the world due to the size of the country and the low population densities across regions [3]. This highlights the increased demand for road transportation in the country and the importance of maintaining a healthy workforce in the trucking industry. However, the nature of the trucking occupation does not reflect this importance. Previous studies suggest fatigue is highly prevalent in truck drivers due to the unique set of risk factors in their working environment, including shift work, prolonged working times, and irregular sleep patterns [4,5]. Perceived levels of physical pain from long periods of sitting or mental discomfort experienced from the isolation of being on the road also contributed to an increased level of fatigue for drivers [6]. Additionally, drivers in the road transportation industry experience a high level of physical and mental problems due to the demands of the driving task. Truck drivers report experiencing a variety of medical issues, such as back problems, high blood pressure, and mental health conditions, which are exacerbated by the extensive distances they cover [7,8]. The health profile of Australian truck drivers is reportedly worse than that of the general population [9], which has been found to lead to increased health and workplace risks, culminating in higher levels of injury and crashes [10].

It is widely acknowledged that fatigue is a serious safety hazard and one of the leading contributing factors to traffic crashes in the road transportation industry [11,12]. Fatigue is defined as a “physical or mental state of a lack of energy and concentration where sleepiness, tiredness, drowsiness, and lethargy are often used interchangeably” [13,14]. Fatigue in the road transport industry is also described as “falling asleep at the wheel” [15]. In the study of Williamsons et al. [16], they described fatigue as “a biological drive for recuperative rest”, indicating that this rest depends on the nature of the fatigue. Sleepiness and mental fatigue are the main forms of fatigue in the transport industry. There is strong evidence of the negative influence of fatigue on driving performance, including decreased judgement ability, slow reaction times, a narrowing of attention, and increased operational errors [17,18]. Worldwide, it was estimated that 10–40% of crashes reported were due to driver fatigue, with a higher percentage of fatal fatigue-related crashes [19,20]. In Australia, nearly 20% of fatal road crashes are caused by driver fatigue, and experiencing fatigue is four times more likely to result in a workplace injury than drug use among truck drivers [21]. The annual cost of fatigue-related road crashes in Australia is around $3 billion, placing a significant burden on the health services and society [22].

To reduce the incidence of traffic crashes and improve truck drivers’ overall health and wellbeing, various studies have sought to understand the complex relationship between the characteristics of this workforce and fatigued driving among truck drivers. Previous studies have identified that long working hours (over 16 h per day) were positively associated with an increased risk of fatigued driving, especially for those who experienced unreasonable waiting or queuing times for loading or unloading [23,24]. A cross-sectional study of truck drivers in Portugal showed that many commercial truck drivers reported excessive daytime sleepiness and impaired sleep duration and quality due to working night shifts [25]. Additionally, an Australian cross-sectional study investigating long-haul truck drivers found trip-based payments (based on kilometres travelled only) were associated with fatigued driving and played a role in extended working hours, distances, and reduced rest time [23,26]. Shifts of more than 8 h and sleep deprivation (less than 8 h of sleep per night) all contributed to fatigue, decreased alertness, and slower reaction times [27]. Lifestyle and behavioural factors common among truck drivers also significantly contribute to fatigue in the trucking industry [28]. For instance, a cross-sectional study of truck and bus drivers in Italy reported that smoking increased the risk of excessive daytime sleepiness due to reduced oxygen in the lungs, brain, and body systems, which exacerbated fatigue [29]. Irregular dietary patterns and unhealthy meal consumption with a high glycaemic index, high sugar, and high fats were also found to be independent risk factors for excessive daytime sleepiness among truck drivers [30].

To date, the impact of long working hours, shift work, trip-based payments, and poor sleep quality on fatigued driving has been commonly reported [31,32]. However, most existing research into truck driver fatigue has been conducted outside of Australia. Current Australian studies of truck drivers focus almost exclusively on the relationship between fatigue and road safety issues such as crashes and violations [33,34], with limited research capturing other factors such as lifestyle-related factors (diet, alcohol, smoking, and physical activity) and pre-existing health conditions. This presents a significant gap in knowledge, as a better understanding of the relationship between these factors will provide critical information for the development of targeted interventions to reduce fatigue.

The aim of this study Is to examine a range of personal, social, and environmental factors associated with fatigue outcomes among Australian truck drivers, including demographic characteristics, occupational characteristics, lifestyle factors, and other health risk factors. The results from this study will be used to inform recommendations for the review and revision of interventions and the development of new initiatives.

## 2. Methods

### 2.1. Study Design

This study was part of the Driving Health Project, which incorporated two cross-sectional studies (an online and telephone survey) and a qualitative study targeted at Australian truck drivers. Segments of data from the 10-min online survey and the 25-min follow-up telephone survey were used in this study. The conceptual model from Crizzle et al. was adopted as an underlying basis for the survey design [35]. The conceptual model illustrates the range of factors that could be associated with health and driving outcomes such as fatigue, including personal, social, and work environment factors.

#### 2.1.1. Recruitment

The detailed description of participant recruitment and data sources for the initial online survey of 1390 Australian truck drivers are described elsewhere [9]. At the end of the initial online survey, eligible participants were invited to take part in a follow-up telephone survey. The Social Research Centre (SRC), a market research consultancy firm, conducted the 25-min telephone survey using computer-assisted telephone interviewing (CATI), with the option to complete it online if preferred. All participants were assured of anonymity and confidentiality and had the right to withdraw at any time.

#### 2.1.2. Participants

The inclusion criteria were drivers aged 18 and over, employed in a job involving the transport of goods in the last 12 months, and able to complete the survey in English. Participants were excluded from the study if they drove vehicles other than trucks or vans.

In total, 471 eligible respondents were contacted following the online survey completion, and 332 completed the follow-up telephone survey (70% response rate).

#### 2.1.3. Outcome Measurements

In the telephone survey, the level of fatigue was measured by two questions drawn from previous surveys [36,37]. The first question asked about the frequency of being fatigued: “How often do you become fatigued while driving for work?” with 5-point Likert scale responses ranging from (0 = never, 1 = less than once a month, 2 = monthly, 3 = weekly, 4 = daily/almost daily). Responses were amalgamated into two categories of fatigued driving risk: “Never”, “Less than once a month”, and “Monthly” were considered low-risk, whilst “Weekly”, and “Daily/Almost daily” were considered high-risk in the consideration of frequency distributions.

The second question asked whether drivers take anything to combat fatigue, with a possible response of “Yes” or “No”. If the answer was “No”, they moved on to the next section. If the answer was “Yes”, they were asked to fill in the question about strategies used to combat fatigue. Commonly used strategies to counteract fatigued driving were categorised into “caffeine or energy drink”, “food or beverage”, “caffeine pills”, and “prescribed medications”. An additional “other” option was also set for drivers to report strategies that were not in the above options. The questions were developed from previous study findings and in consultation with truck drivers.

### 2.2. Risk Factors and Measurements

Determinants of fatigue included in this study were identified and selected based on the conceptual model of the Driving Health study [38], including personal factors, occupational factors, workplace environment factors, regulatory factors, and lifestyle factors. A summary of the factors is presented in Table 1 and described below. For a detailed description of the determinants captured in the telephone survey and how these were collapsed, please refer to Appendix A.

Personal factors: Personal factors included drivers’ basic demographic characteristics, pre-existing physical and mental health conditions, and financial stress. Physical health was measured by self-reported general health questions adopted from the National Health Survey [39]. Mental health was determined from self-reported conditions by participants. Participants’ financial stress question was derived from the National Health Survey 2014/15, which asked about financial position with a response of “Yes” or “No” [40].

Occupational factors: Occupational factors reflected drivers’ work characteristics and working conditions, including shift type (day or night), payment type (per kilometre travelled or per hour), working hours, work tasks, work time, schedules, and payment for delays. The latter four were measured by adapting the Occupational Health and Safety (OHS) vulnerability measurements developed by the Institute for Work & Health, using a 5-point Likert scale ranging from (0 = never, 1 = less than once a month, 2 = monthly, 3 = weekly, 4 = daily/almost daily) [41].

Workplace environment factors: These refer to the physical work environment and workplace violence, i.e., how drivers were treated by management and the public. The physical work environment was measured by 7 items adapted from the OHS Vulnerability Scale, with responses ranging from ‘Never’ to ‘Daily/Almost daily’ [41]. An example question is “In the last 12 months, how often in your job did you experience discomfort by mechanical vibration or shock in your work?”. Four questions were used to explore drivers’ experience of workplace violence. An example question is “In the past 12 months, have you been verbally abused in your workplace” answered with (0 = no, 1 = yes).

Regulatory factors: refers to breaking regulation behaviours where four questions were used to assess the participant’s behaviour, using a 5-point Likert scale from (0 = never, 1 = less than once a month, 2 = monthly, 3 = weekly, 4 = daily/almost daily). A sample item is, “In the last 12 months, how often in your job did you drive in excess of the speed limit?”. Fatigue management was assessed by asking drivers about their experience receiving fatigue management training, who answered with (0 = no fatigue management, 1 = basic fatigue management, 2 = advanced fatigue management, or 3 = other).

Lifestyle factors: These included information regarding drivers’ self-reported diet, alcohol consumption, smoking, and level of physical activity. Diet questions were adopted from the National Health Survey (NHS) Module 13-Dietary behaviours, assessing serves of vegetables and fruit per day or per week [42]. Drivers’ self-reported intake was compared with current recommended guidelines of two servings of fruit and five to six servings of vegetables per day [42]. Physical activity questions were adopted from the NHS Module 10-Exercise, assessing the hours or minutes per week, defined as moderate (working hard enough to raise your heart rate and break a sweat) or vigorous exercise (not being able to say more than a few words without pausing for a breath) [40]. Whether a person smoked (including e-cigarettes) or not, they were described as smokers or non-smokers. Alcohol consumption questions were selected from three AUDIT-C 3 screening questions to identify the risk of alcohol misuse [43]. (1) How often do you have a drink containing alcohol? (2) How many drinks containing alcohol do you have on a typical day when you are drinking? (3) How often do you have six or more drinks on one occasion ? Points were allocated to each response following the AUDIT-C 3 scoring system (0–12 points) to determine a risk of problem alcohol use score. A score of ≥4 was defined as “at risk” of alcohol misuse.

### 2.3. Statistical Analysis

The original online and telephone survey responses of participants were extracted and combined in the same database using an unidentifiable study ID gendered by the data manager. Risk factor variables were collapsed and recoded using STATA (16.1, StataCorp LLC, College Station, TX, USA), an integrated statistical software package. Cronbach’s alpha reliability test indicated high reliability across measurements with multiple Likert questions and a satisfactory alpha value of 0.826. The number of missing and/or “prefer not to say” responses were assessed and was less than 3% for each item. Responses with missing outcome variables were excluded from the final analysis.

Descriptive analysis was used to report the frequency of risk factors and levels of fatigue. Logistic regression was used to investigate the association between fatigue outcomes (high-risk of fatigued driving; using stimulants to combat fatigue) and risk factors. Fatigued driving was coded as a binary variable with a value of 1 representing a high-risk of fatigued driving (drivers who experienced fatigued driving “Weekly”, and “Daily/Almost daily”) and a value of zero representing a low-risk of fatigued driving (drivers who experienced fatigued driving “Never”, “Less than once a month” or “Monthly”). A univariate regression was first performed, and variables with a *p*-value of less than 0.2 were included for further multivariable analysis. A similar analysis was performed to determine the association of factors that predicted strategies to combat fatigue. A value of 1 represented using any strategies to combat fatigue, and a value of zero represented no strategies used to combat fatigue. In this study, statistical significance was set at a *p*-value less than 0.05.

## 3. Results

### 3.1. Baseline Characteristics

In total, 338 valid online surveys with the corresponding follow-up telephone surveys were collected. Six participants withdrew from the study, leaving a total of 332 surveys for data analysis. Table 2 provides the demographic characteristics of the participants. Most of the drivers were male (98%), with ages ranging from 18 to 65 years. Almost one third of drivers were >55 years old, which represented 34.3% of the sample, and 19.6% were less than 35 years old. The majority of participants were employee drivers (84.6%), and nearly two-thirds (59.8%) drove short-haul (<500 km daily). Over half the drivers (51.5%) had more than 20 years of driving experience, and two thirds (63.3%) were educated at a level above high school. The majority of drivers (75.6%) were in some form of relationship, with 39.6% having partners and children. Drivers report a lower level of financial stress (72.7%), when compared to general Australian households [44]. Drivers rated their general health as good (35.2%) and poor to fair (35.0%), with more than half of truck drivers (55.8%) reported as obese (BMI > 29).

Figure 1 shows the distribution of drivers’ responses to fatigue measurement questions, with experiencing fatigue while working being commonly reported. The majority of drivers (62.0%) in this study reported experiencing fatigue and falling asleep at the wheel at work almost every month or every week (Figure 1a). Figure 1b shows basic fatigue management training was the most common fatigue management programme completed by drivers (53.4%), with only 7.7% of drivers reporting advanced fatigue management training. Over a third of drivers reported having no accreditation fatigue management training (37.4%). Furthermore, caffeine or energy drinks were the most common ways to combat fatigue, followed by food or beverages. Drivers were less likely to use caffeine pills and/or prescribed medication to combat fatigue (Figure 1c).

### 3.2. Factors Associated with Being at High-Risk of Fatigued Driving

Table 3 describes the adjusted odds ratios (ORs) of risk factors for the outcome of being at high risk of fatigued driving. The unadjusted ORs of univariable analysis are provided in Appendix A. Longer working hours, heavy work tasks, breaking regulations, poor sleep, and feelings of loneliness were found to be positively associated with being at high risk of fatigued driving. The regression analysis was run on fatigue management training, but none of the factors were significantly associated with the outcome. The odds of being in the high risk group for fatigued driving decreased with age but were not statistically significant. Drivers who were not partnered with dependent children (OR: 0.14, 95% CI: 0.04–0.50; *p* = 0.002) and partnered with dependent children (OR: 0.336; 95% CI: 0.131–0.860; *p* = 0.023) had significantly lower odds of being at high risk of fatigue compared to those without a partner and without dependent children.

There was a clear positive association between fatigue and longer working hours. The odds of being in the high-risk group for fatigued driving were nearly three times higher in drivers who worked 40–60 h compared to those who worked < 40 h (OR: 2.96, 95% CI: 1.27–6.97, *p* = 0.013). The odds were also higher for drivers working > 60 h per week, but this was not significant. Drivers who were categorised in the moderate-risk work task group had nearly double the odds of high-risk fatigued driving than those in the low-risk group (OR: 1.97, 95% CI: 1.04–3.72, *p* = 0.036). Drivers who exhibited breaking regulation behaviour had twice the odds of experiencing high-risk fatigued driving (OR: 2.11, 95% CI: 1.17–3.80, *p* = 0.013) than those who reported no breaking regulation behaviour.

Sleep showed a strong association with fatigued driving. In comparison to the drivers in the low-risk of poor sleep group, drivers categorised as high-risk of poor sleep had seven times (95% CI: 2.26–21.67, *p* = 0.001) higher odds of being at high-risk of fatigued driving at work. Drivers who reported experiencing loneliness also had double the odds of being at high risk of fatigued driving compared to those who reported never or rarely feeling loneliness (OR: 2.16, 95% CI: 1.64–3.99, *p* = 0.015).

### 3.3. Factors Influencing the Choice of Strategies Used to Combat Fatigue

Age, work environment, and smoking were found to be associated with using stimulants to combat fatigue (Table 4). Drivers aged between 45 and 54 years (OR: 0.32; 95% CI: 0.14–0.75) and those over 55 years (OR: 0.28; 95% CI: 0.12–0.68) had significantly lower odds of using stimulants to combat fatigue compared to younger drivers. Drivers wiinth high-risk work environments had three times the odds of using stimulants to combat fatigue than those in low-risk work environments (OR: 3.04, 95% CI: 1.61–7.95, *p* = 0.024). Furthermore, smokers also showed higher odds of using stimulants to combat fatigue compared to non-smokers (OR = 2.43; 95% CI: 1.24–4.75). The unadjusted ORs of univariable analysis are provided in Appendix A.

## 4. Discussion

The aim of this study was to examine the relationship between personal, social, and environmental factors and fatigue outcomes among Australian truck drivers. Our study suggested that one in two drivers reported experiencing fatigue and falling asleep at the wheel at work almost every month or every week. Prolonged working hours, high-risk work tasks, poor sleep, and a feeling of loneliness were found to be significantly associated with high-risk fatigued driving. The results of this study provide critical insight into some of the determinants of fatigue, which is important in the review and revision of interventions to reduce fatigue and, ultimately, reduce the risk of vehicle incidents.

In this cross-sectional study, regression analyses suggest that long working hours are a key occupational factor contributing to fatigued driving. As in our previous report, nearly 40% of drivers can be on duty for over 60 h a week [9], which is 58% higher than the Australian standard working week hours. These findings are consistent with current literature, which states that truck drivers who drive longer distances and spend more time driving have higher odds of falling asleep at the wheel [23,25,29]. In Australia, standard guidelines recommend that a driver should not work more than 12 h and have at least seven continuous hours of stationary time within a 24-h period [45]. However, with the current high demand for the logistics market, the number of truckers working longer and more demanding hours is on the rise, making it hard for truck drivers to meet this requirement. Several studies have shown the harmful impact of long working hours on the health and wellbeing of truck drivers, including increased risk of obesity, diabetes, and cardiovascular disease [46,47]. In addition, mental health problems such as anxiety and depression are also positively associated with prolonged working hours (over 60 h per week) in the industry [48,49].

Poor sleep was also found to be a significant predictor of being at high risk of fatigued driving at work. In line with previous studies on sleep disorders in truck drivers, sleep deprivation was positively associated with drowsiness and excessive somnolence in the drivers [24,26]. Sleep deprivation seriously affects a driver’s attention, judgment, decision-making, coordination, alertness, and reaction times [50,51]. These risks are especially harmful for truck drivers, who consistently spend more time on the road than other motorists. Previous studies exploring sleep and vehicle crashes found that the risk of experiencing sleep-related crashes was higher on monotonous highways than on city roads [52,53]. Therefore, primary prevention is an unavoidable step to deal with sleep problems in the trucking industry.

A common way to combat fatigued driving in our study was the use of “caffeine or energy drink” and “food or beverage”. This is consistent with an American study, which found drinking caffeinated beverages, stopping for a nap, and chewing ice could prevent a road incident [54]. Our findings also reveal that younger drivers were more likely to use stimulants to combat fatigue during work [55].

Truck drivers are commonly exposed to repetitive movements of the hands and wrist (e.g., use of the steering wheel and gears), manual lifting of heavy items, and interacting with hazardous substances [56,57]. In our study, drivers routinely exposed to these high-risk work tasks had increased odds of being at high risk of fatigued driving. Tasks like working with hazardous substances and working in environments with high noise levels require concentration and vigilance, which may limit the opportunity to “switch off”, which in turn may exacerbate fatigue [38]. It should be the responsibility of management to address modifiable high-risk work tasks like drivers standing for long periods of time (when waiting for loads), working in bent postures (packing/unpacking), or performing tasks they are unfamiliar with. Improving physical work conditions and therefore reducing the frequency with which drivers are exposed to these high-risk work tasks could minimise the risk of drivers experiencing fatigue behind the wheel.

We found that drivers with dependent children had lower odds of fatigued driving during work compared to those without dependent children. Supportive family relationships provide emotional care and can encourage good health habits like a good sleep routine [58]. Family members can also help people cope with stressful events, thereby reducing potential risk factors that disrupt sleep quality [59]. This is further supported by our findings on the relationship between lonely and fatigued driving, where drivers who reported feeling loneliness had twice the odds of fatigued driving.

The Driving Health study concluded that social connections and support were crucial to truck driver health and wellbeing [7]. It is widely acknowledged that truck driving is a lonely occupation, and drivers often have to travel for long periods of time by themselves due to work obligations, compared to other professional drivers who are in an environment with passengers [25]. Current evidence shows that loneliness and social isolation have a significant impact on people’s physical and mental health, notably increasing the rates of depression and anxiety [60,61]. Further analyses confirmed the association between experiencing loneliness and elevated levels of psychological distress in this population [38]. Therefore, encouraging drivers to maintain contact with others is not only essential to improve drivers’ health and wellbeing but also important for safety on the road by combating fatigue. Future strategies could include planning to stay in touch with family members during rest breaks, talking to other truck drivers on social media, and staying active in the trucking community.

### Implication

The findings of this study inform the recommendations to reduce fatigue of drivers, and ultimately, improve the safety of this critical cohort of the workforce. The results suggest a range of systemic issues influencing fatigue outcomes, which suggests that intervention focused solely on changing the lifestyle and individual attributes of drivers is unlikely to create any significant change. The results suggest a system-level review and revision of rostering schedules to manage working hours, wellbeing initiatives to promote mental and physical health, and strategies to promote a sense of belonging in the workplace are all required. A review and revision of regulatory strategies focused on driving and rest times and the provision of truck stops are also supported.

## 5. Strengths and Limitations

The strengths of this study included using data collected from one of the largest Australian national surveys of truck drivers. Participants were from all Australian states and included both short- and long-haul truck drivers who drove a wide range of trucks across various experience levels. The population completing the online survey was found to be representative of truck drivers in Australia [6], and the characteristics of drivers completing the telephone survey were representative of the online survey [37]. Secondly, the surveys captured a wide range of factors that are associated with driving fatigue, including personal, occupational, lifestyle, and health risk factors.

However, there are several limitations that should be acknowledged for this study. First, the surveys relied on self-reported responses and the influence of driver memory, which is likely to generate recall, measurement, or information biases. Second, although the online survey was anonymous, the subsequent telephone survey was not, which may have influenced responses to sensitive topics like alcohol consumption, near misses, or injuries. Third, the average age of the sample size is older than the national average in the trucking industry and may not be a fair representation of all ages. Lastly, the survey was designed to explore a wide range of determinants of truck driver health and driving outcomes, rather than those specific only to fatigue. Therefore, some validated fatigue measurement tools and questionnaires adjusted for fatigue were not used in the surveys, resulting in limited comparison across other literature.

## 6. Conclusions

Our study found that prolonged working hours, high-risk work tasks, poor sleep, and feelings of loneliness were positively associated with fatigued driving. In contrast, having dependents was identified as a protective factor against fatigued driving. The findings presented in this study demonstrate the need for improved management of truck driver working hours within the constraints of regulatory and occupational requirements. Wherever possible, companies and managers should encourage drivers to stay socially connected with their family as well as peers in an effort to reduce the risk of fatigued driving, improve driver wellbeing, and ultimately create a safer work environment for this group.

## Figures and Tables

**Figure 1 ijerph-20-02732-f001:**
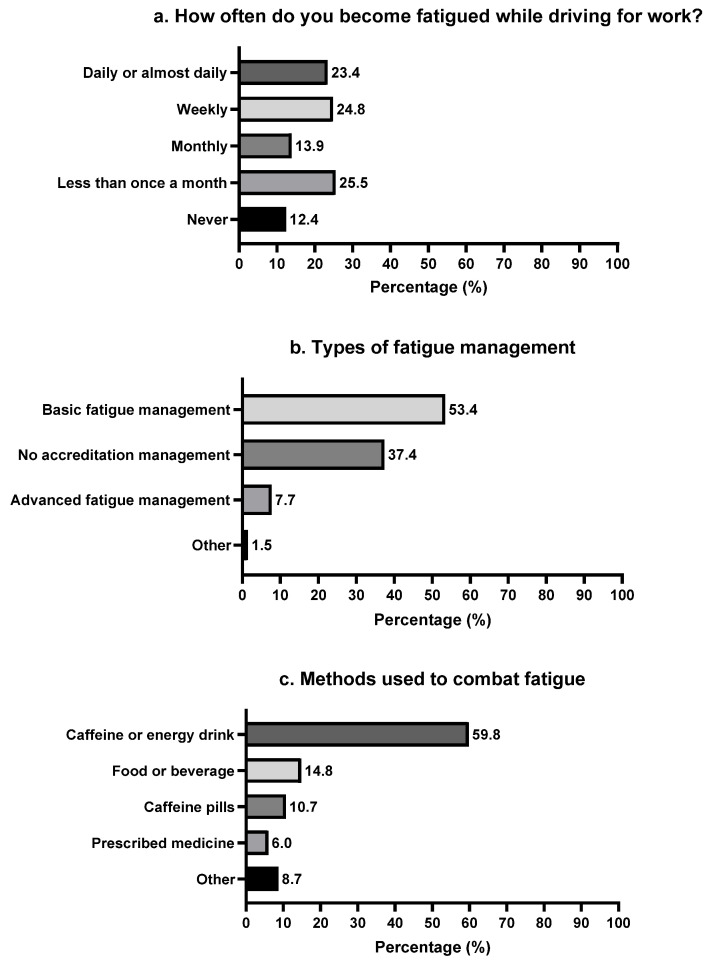
Distribution of participants’ responses to fatigue measurement questions: (**a**) Frequency of being fatigued; (**b**) Experiencing in having fatigue management training; (**c**) Methods used to combat fatigue.

**Table 1 ijerph-20-02732-t001:** Determinants of fatigue included in this study.

Personal Factors	Age ^#^, family situation *, educational status *, mental health ^#^, physical health ^#^, financial stress *
Occupational factors	Shift type ^#^, payment type ^#^, working hours ^#^, work tasks *, work time and schedule *, payment for delays *
Work environment factors	Work environment *, workplace violence *
Regulatory factors	Fatigue management *, breaking regulation behaviour *
Lifestyle factors	Diet *, alcohol *, smoking *, physical activity *
Health risk factors	Sleep *, fatigue *, pain ^#^, medication use *, loneliness *, BMI ^#^.

^#^ from the online survey, * from the telephone survey.

**Table 2 ijerph-20-02732-t002:** Demographic characteristics of survey respondents.

Variables	All *n* = 332 (%)
Age (years)	
<35 years	65 (19.6)
35–44 years	67 (20.2)
45–54 years	86 (25.9)
>55 years	114 (34.3)
Gender	
Male	323 (97.9)
Female	7 (2.1)
Employment status	
Owner driver	51 (15.4)
Employee driver	280 (84.6)
Driver types	
Short-haul driver (<500 km)	198 (59.8)
Long-haul driver (>500 km)	133 (40.2)
Driving experience	
<5 years	45 (13.6)
5–20 years	116 (34.9)
>20 years	171 (51.5)
Education status (highest level of education)	
Above high school	209 (63.3)
Completed high school and lower	121(36.7)
Family status	
No partner with no dependent children	52 (15.9)
No partner with dependent children	28 (8.5)
Partner with no dependent children	118 (36.0)
Partner with dependent children	130 (39.6)
Financial stress status	
Low	240 (72.7)
High	90 (27.3)
BMI (Body mass index)	
Under or normal weight (BMI < 25)	59 (18.0)
Overweight (BMI 25–29)	86 (26.2)
Obese (BMI > 29)	183 (55.8)
General health	
Very good	99 (29.8)
Good	117 (35.2)
Poor to fair	116 (35.0)

BMI was calculated based on self-reported height and weight.

**Table 3 ijerph-20-02732-t003:** Adjusted multivariate logistic regression on high-risk of fatigued driving.

High-Risk Fatigue	Odds Ratio	95% Confidence Interval	*p*-Value
Personal factors			
Age			
<35 years	Ref		
35–44 years	0.778	0.311–1.949	0.592
45–54 years	0.532	0.222–1.274	0.157
>55 years	0.551	0.225–1.352	0.193
Family status			
No partner with no dependent children	Ref		
No partner with dependent children	0.140	0.039– 0.495	**0.002**
Partner with no dependent children	0.674	0.260–1.746	0.416
Partner with dependent children	0.336	0.131–0.860	**0.023**
Occupational factors			
Working hours			
<41 h	Ref		
41–60 h	2.960	1.256–6.974	**0.013**
>60 h	2.412	0.971–5.990	0.058
Work task			
Low-risk	Ref		
Moderate-risk	1.970	1.044–3.717	**0.036**
High-risk	1.457	0.614–3.456	0.394
Work time and schedule			
Low-risk	Ref		
High-risk	1.290	0.726–2.296	0.385
Environmental factors			
Work environment			
Low-risk	Ref		
High-risk	1.432	0.729–2.813	0.297
Workplace violence			
No	Ref		
Yes	0.746	0.400–1.389	0.355
Regulatory factors			
Breaking rules and regulations			
No	Ref		
Yes	2.11	1.172–3.799	**0.013**
Fatigue management			
No fatigue management	Ref		
Basic/advanced fatigue management	0.884	0.498–1.571	0.675
Lifestyle factors and health risks			
Diet			
Does not meet the guidelines	Ref		
Meets the guidelines	0.871	0.491–1.547	0.638
Exercise			
Does not meet the guidelines	Ref		
Meets the guidelines	0.654	0.366–1.168	0.151
Smoking			
Non-smokers	Ref		
Smokers	1.607	0.794–3.250	0.187
Sleep			
Low-risk	Ref		
High-risk	6.990	2.255–21.665	**0.001**
Pain			
No	Ref		
Yes	1.629	0.908–2.921	0.102
Loneliness			
No	Ref		
Yes	2.155	1.164–3.990	**0.015**

Ref = reference. Bold values denote statistical significance at the *p* < 0.05 level.

**Table 4 ijerph-20-02732-t004:** Factors associated with having used something to combat fatigue.

Combating Fatigue—Yes	Odds Ratio	95% Confidence Interval	*p*-Value
Personal factors			
Age category			
<35 years	Ref		
35–44 years	0.883	0.387–2.016	0.768
45–54 years	0.303	0.133–0.691	**0.005**
>55 years	0.303	0.133–0.695	**0.005**
Education level			
Above high school	Ref		
High school and lower	0.894	0.482–1.657	0.722
Financial stress			
Low	Ref		
High	1.627	0.875–3.024	0.124
Occupational factors			
Payment type			
Flat rate	Ref		
Per trip/delivery	2.149	0.796–5.805	0.131
Single-time pay	2.139	0.941–4.862	0.070
Kilometre rate	1.886	0.796–4.467	0.149
Other	2.676	0.967–7.406	0.058
Work time and schedule			
Low-risk	Ref		
High-risk	0.903	0.466–1.751	0.763
Work-task			
Low-risk	Ref		
Moderate-risk	1.912	0.898–4.068	0.093
High-risk	1.152	0.453–2.931	0.766
Environmental factors			
Workplace violence			
No	Ref		
Yes	1.527	0.819–2.846	0.183
Regulatory factors			
Breaking rules and regulations			
No	Ref		
Yes	1.300	0.671–2.522	0.437
Lifestyle factors and health risks			
Exercise			
Does not meet the guidelines	Ref		
Meets the guidelines	0.676	0.378–1.210	0.187
Smoking			
Non-smoker	Ref		
Smoker	2.468	1.282–4.751	**0.007**
Drinking			
Light drinker	Ref		
Heavy drinker	1.615	0.897–2.909	0.110
Sleep			
Low-risk	Ref		
High-risk	1.454	0.696–3.041	0.320
Pain			
No	Ref		
Yes	1.243	0.641–2.411	0.519
Loneliness			
No	Ref		
Yes	1.077	0.586–1.982	0.811

Ref = reference. Bold values denote statistical significance at the *p* < 0.05 level.

## Data Availability

Due to privacy and ethical concerns, the data cannot be made available to the public.

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
