# Peer review of "Factors Associated with Fatigued Driving among Australian Truck Drivers: A Cross-Sectional Study"

_ijerph, 2023, doi:10.3390/ijerph20032732_

Round 1
Reviewer 1 Report
Paper Review
Manuscript ID: ijerph-2193626
The manuscript (MS) was written fairly well, although there are a number of improvements that
can be made and incorporated into the MS.
Abstract:
1. Suggest modifying ‘Fatigue driving is the leading...’ to ‘one of the leading...’.
2. Limited research. Suggest adding ‘limited research in Australia(?)’
3. “Loneliness with fatigue”. How about saying the social aspects (such as lonliness...)?
4. Conclusions: Suggest referring to only study findings, and suggest potential
interventions (pertaining to the findings)
Introduction
1. Suggest adding the work of Mabry et al. (2022) for the background.
2. ‘... a leading contributing factor...’ (line 47) into ‘... one of the leading contributing
factors...’
3. Suggest adding Williamson et al. (2011) for fatigue concept and definition
4. ‘..isolation of being on the road...’ (line 56) is demonstrated in Wilder et al. (1994)? (Ref#
15)
5. The paragraph (line 58-67) lacks unity. Please revise. Move some of the sentences to the
early part of the introduction (background)
6. Check each paragraph. Ensure unity (e.g., line 46-57).
7. Research done in Australia. Line 97. Was this actually done in India?
Method
1. It would be nice if the study can compare the number of respondents to other studies
2. Driving Health study. Need citation
3. Citation for NHS Module 13 (line 200)
4. What is STATA 16? (Some readers may not know this statistical software)
5. Please explain (with citations) how high-risk of fatigue is determined. Any citation?
Discussion
1. Please state major findings explicitly in the first paragraph
2. Reference # 48 is not related to the statement. Please remove or substitute.
3. Please double check References # 49 and 50. Are they related to the transportation
sector?
4. For the 2nd paragraph, please address hours of service in other industrialized nations.
What causes drivers to work longer hours in these studies?
5. Please check citation (line 363)
6. Is driving task really considered as repetitive tasks? (Line 369). Provide citation.
7. Third paragraph should only address sleep deprivation and the mechanisms associated
with fatigue.
8. Loneliness was stated as a contributing factor. The proposed countermeasures,
however, seem trivial and need to be elaborated further (with citations).
9. Last paragraph seems to be out of context. Please reconsider. If this paragraph is
intended to address intervention strategies, please provide those that pertain to major
findings.
Author Response
Abstract
- Suggest modifying ‘Fatigue driving is the leading...’ to ‘one of the leading...’.
Re: The sentence has been revised to “Fatigued driving is one of the leading factor contributing to road crashes in the trucking industry”
- Limited research. Suggest adding ‘limited research in Australia(?)’
Re: The sentence has been revised to “there is limited research in Australia investigating the impact of demographic, occupational or lifestyle factors on fatigue.”
- “Loneliness with fatigue”. How about saying the social aspects (such as loneliness...)?
Re: We revised the sentence to “The increased risk of fatigue in truck drivers is associated with prolonged working hours, poor sleep, and the social aspects such as loneliness.”
- Conclusions: Suggest referring to only study findings, and suggest potential interventions (pertaining to the findings)
Re: We thank reviewer for this comment. We revised the conclusion to “The increased risk of fatigue in truck drivers is associated with prolonged working hours, poor sleep, and the social aspects such as loneliness). Further interventions seeking to reduce driver fatigue should consider the impact of work schedules, availability of quality sleeping spaces and level of social connections.”
Introduction
- Suggest adding the work of Mabry et al. (2022) for the background.
Re: We thank reviewer for this suggestion. Mabry’s work has been referenced in the introduction section.
- ‘... a leading contributing factor...’ (line 47) into ‘... one of the leading contributing
factors...’
Re: The sentence has been modified accordingly.
- Suggest adding Williamson et al. (2011) for fatigue concept and definition.
Re: Fatigue concept and definition have been added with citation from Williamson et al. (2011).
New reference has been added in the reference list. “ In the study of Williamsons et al. (2011), they described fatigue as “a biological drive for recuperative rest”, indicating this rest depends on the nature of the fatigue. Sleepiness and mental fatigue are the main forms of fatigue in the transport industry.”
- ‘..isolation of being on the road...’ (line 56) is demonstrated in Wilder et al. (1994)?
Re: We thank reviewer for picking this up. The incorrect reference (Ref#15) has been replaced with the correct one in the reference list.
- The paragraph (line 58-67) lacks unity. Please revise. Move some of the sentences to the early part of the introduction (background)
Re: This paragraph was revised and moved into the pervious paragraph.
- Check each paragraph. Ensure unity (e.g., line 46-57).
Re: The first two paragraphs in 1.1 Background have been restructured for unity.
- Research done in Australia. Line 97. Was this actually done in India?
Re: We thank reviewer for picking this up. The incorrect reference (Ref#34) has been replaced with the correct one in the reference list.
Method
- It would be nice if the study can compare the number of respondents to other studies
Re: This study included using data collected from one of the largest Australian national surveys of truck drivers. This was acknowledged in the Strength and Limitation section.
- Driving Health study. Need citation
Re: Citation has been added.
- Citation for NHS Module 13 (line 200)
Re: Citation has been added.
- What is STATA 16? (Some readers may not know this statistical software)
Re: In the revised manuscript, we have provided the definition of STATA:
“Risk factor variables were collapsed and recoded using STATA (16.1, StataCorp LLC, College Station, TX), an integrated statistical software package.”
- Please explain (with citations) how high-risk of fatigue is determined. Any citation?
Re: High-risk of fatigue in this study is defined as drivers who experienced fatigue driving “Weekly”, and “Daily/Almost daily” in this study in the consideration of the frequency distributions of 5-point Likert scale. We have made an explanation in the revised manuscript.
Discussion
- Please state major findings explicitly in the first paragraph
Re: The major findings have been highlighted in the first paragraph:
“Our study suggested that two in three drivers reported experiencing fatigue and falling asleep at the wheel at work almost every month or every week. Prolonged working hours, high-risk work tasks, poor sleep and feeling of loneliness were found to be significantly associated with high-risk fatigued driving.”
- Reference # 48 is not related to the statement. Please remove or substitute.
Re: The reference has been double-checked and corrected.
- Please double check References # 49 and 50. Are they related to the transportation
sector?
Re: The reference has been double-checked and corrected.
- For the 2nd paragraph, please address hours of service in other industrialized nations.
Re: We have provided more information regarding the Australian standard working hours for a week:
“As in our previous report, nearly 40% of drivers can be on duty for over 60 hours a week, which is 58% higher than the Australian standard working week hours.”
- What causes drivers to work longer hours in these studies?
Re: We added new explanation in this paragraph:
“However, with the current high demand for the logistics market, the number of truckers working longer and more demanding hours is on the rise, making the truck drivers hard to meet this requirement.”
- Please check citation (line 363)
Re: The citation has been corrected in the revised manuscript.
- Is driving task really considered as repetitive tasks? (Line 369). Provide citation.
Re: In the revised manuscript, references have been provided to support this statement.
- Third paragraph should only address sleep deprivation and the mechanisms associated with fatigue.
Re: We thank the reviewer for this comment. We deleted the last sentence and added a conclusion.
- Loneliness was stated as a contributing factor. The proposed countermeasures, however, seem trivial and need to be elaborated further (with citations).
Re: We thank the reviewer for this comment. In the revised manuscript, this paragraph has been condensed by deleting trivial content.
- Last paragraph seems to be out of context. Please reconsider. If this paragraph is
intended to address intervention strategies, please provide those that pertain to major findings.
Re: We thank the reviewer for this comment. We removed this paragraph in the revised manuscript but add an implication section.

Reviewer 2 Report
Fatigued driving among Australian truck drivers: a cross-sectional study
This is a relevant and interesting contribution focused on the role of demographic, occupational, lifestyle, and other health risk factors associated with fatigue among Australian truck drivers. I believe that this focus should be reflected on the title of the article to provide more clarity. Also, a few other suggestions would improve the overall chances of it being published:
1. Please adhere to the journal’s reference style.
2. Objectives should be more thoroughly described to display all methodological procedures used.
3. Shouldn’t table 1 (2.5) be considered in the results section?
4. What is the mean age (SD) of participants?
5. Some of the instruments used are susceptible of being analyzed with reliability analyses. Please provide Cronbach’s alphas.
6. Authors should provide an implications section.
Best wished.
Author Response
This is a relevant and interesting contribution focused on the role of demographic, occupational, lifestyle, and other health risk factors associated with fatigue among Australian truck drivers. I believe that this focus should be reflected on the title of the article to provide more clarity. Also, a few other suggestions would improve the overall chances of it being published:
- Please adhere to the journal’s reference style.
Re: We thank you for this comment. The reference has been formatted in adhering to the journal’s requirements.
- Objectives should be more thoroughly described to display all methodological procedures used.
Re: We revised the objective to “This cross-sectional study examines the role of demographic, occupational, lifestyle, and other health risk factors associated with fatigue driving using national survey data of Australian truck drivers.”
- Shouldn’t table 1 (2.5) be considered in the results section?
Re: Table1 presents the determinants we selected as potential factors that may be associated with the outcome. Therefore, it was included in the method section.
- What is the mean age (SD) of participants?
Re: In the online survey, participants were asked to choose their age category (10 years band). Therefore, the mean age could not be estimated in this study.
- Some of the instruments used are susceptible of being analyzed with reliability analyses. Please provide Cronbach’s alphas.
Re: The Cronbach’s alpha reliability test was performed on measurements that has multiple Likert questions. The results indicated high reliability with a satisfactory alpha value of .826. This is further explained in the revised manuscript.
- Authors should provide an implications section.
Re: An implication section has been added to the revised manuscript.
